# Driving Attention State Detection Based on GRU-EEGNet

**DOI:** 10.3390/s24165086

**Published:** 2024-08-07

**Authors:** Xiaoli Wu, Changcheng Shi, Lirong Yan

**Affiliations:** 1College of Physics and Electronic Engineering, Hanjiang Normal University, Shiyan 442000, China; wuxiaoli@hjnu.edu.cn; 2Hubei Key Laboratory of Advanced Technology for Automotive Components, Wuhan University of Technology, Wuhan 430070, China; lirong.yan@whut.edu.cn

**Keywords:** driving distraction, EEG, SVM, EEGNet, GRU-EEGNet

## Abstract

The present study utilizes the significant differences in θ, α, and β band power spectra observed in electroencephalograms (EEGs) during distracted versus focused driving. Three subtasks, visual distraction, auditory distraction, and cognitive distraction, were designed to appear randomly during driving simulations. The θ, α, and β band power spectra of the EEG signals of the four driving attention states were extracted, and SVM, EEGNet, and GRU-EEGNet models were employed for the detection of the driving attention states, respectively. Online experiments were conducted. The extraction of the θ, α, and β band power spectrum features of the EEG signals was found to be a more effective method than the extraction of the power spectrum features of the whole EEG signals for the detection of driving attention states. The driving attention state detection accuracy of the proposed GRU-EEGNet model is improved by 6.3% and 12.8% over the EEGNet model and PSD_SVM method, respectively. The EEG decoding method combining EEG features and an improved deep learning algorithm, which effectively improves the driving attention state detection accuracy, was manually and preliminarily selected based on the results of existing studies.

## 1. Introduction

The act of driving is a complex task that requires the coordinated use of a range of cognitive skills, including planning, memory, motor control, and visual abilities. These abilities vary considerably from one individual to another and depend on cognitive skills and attention levels [1]. Distracted attention reduces the driver’s level of perception and decision-making ability, negatively affects vehicle control, and is a significant factor in a large number of accidents [2]. The National Highway Traffic Safety Administration (NHTSA) reports that a total of 39,221 fatal crashes occurred in the United States in FY2022, involving 60,048 drivers and resulting in 42,514 deaths. Of these, 3047 crashes were related to driver distraction, accounting for 8% of all fatalities [3]. Several studies have demonstrated that distraction while driving impairs driving performance [4,5,6,7,8,9,10,11,12,13]. If driving distraction can be detected in a timely and effective manner and interventions can be made, the degree of injury can be reduced, or the accident may even be avoided. Therefore, the research on driving distraction detection methods is of great significance to improving driving safety.

There are five common methods for detecting driving distractions: performance-based, driver body posture, driver physiological characteristics, subjective report, and hybrid detection [14]. However, all of these methods have certain limitations; it is difficult to distinguish between types of distraction in driving performance-based methods, the use of a camera to analyze the driver’s body posture to detect distraction is susceptible to light (e.g., daytime and nighttime), the subjective report is subject to subjectivity, and the mixed measurements suffer from the problem of difficulty in synchronizing the data from multiple data sources. EEGs, on the other hand, result from a series of drivers’ mental activities and are characterized by objective accuracy, high temporal resolution, and the ability to detect different types of distraction. In particular, compared with other methods that can only detect distraction when it is already underway or after it has occurred, EEG technology can detect when the distraction has just occurred. This is because physiological changes begin when the driver is in the initial stages of distraction. Therefore, driving distraction detection using EEG technology is an effective means to improve driving safety.

Differences in the characteristics of EEG signals produced during drivers’ different driving attention states were found and used to detect distraction. EEG signals’ theta (4–8 Hz), alpha (8–13 Hz), and beta (13–35 Hz) activities during driving distraction are significantly different from those during focused driving [15]. And this difference varies for different types of distraction. An EEG signal’s alpha spindle rate was significantly higher during auditory distraction than during focused driving without a subtask [16]. Right parietal alpha and beta activity were negatively correlated with the degree of visual distraction [17]. Alpha and theta band power were positively correlated with cognitive load [18]. This difference has also been demonstrated using neuroimaging to study driving attention states and brain region activation; significant activation was found in the right frontal cortical regions of drivers during distraction [19,20].

EEG technology has been widely used for the detection of distracted driving, such as visual distraction [17,21], auditory distraction [16,22], and cognitive distraction [23,24,25], due to its ease of portability, insensitivity to head motion, and high temporal resolution. However, EEG also has the characteristic of a low signal-to-noise ratio due to the signal being susceptible to environmental noise, as well as physiological activity and motion artifacts, which are difficult to eliminate, resulting in poor accuracy of distraction detection. Therefore, current research focuses on EEG decoding algorithms to improve the accuracy of distraction detection. Some studies used traditional machine learning algorithms for EEG decoding, such as singular value decomposition (SVD) to extract the features of each frequency band of the EEG signal, followed by SVM to classify them to achieve the detection of the driver’s cognitive state, which achieved a detection accuracy of 96.78% [26]. Other studies used deep learning methods for decoding; for example, Yan et al. [27] used EEGNet combined with LSTM to detect visual and auditory distraction, and obtained an average accuracy of 71.1%. Li et al. [28] used a CNN combined with a GRU (gated recurrent unit) for dichotomous classification (distracted vs. non-distracted); the resulting accuracy was between 82% and 100%, and the accuracy of the triple classification (mobile phone operation distraction, two-back distraction, and non-distraction) was between 76% and 98%. Wang et al. [29] performed driver cognitive load detection by extracting the EEG signal’s θ, α, β, and γ band power spectra by using a neural network and an SVM, respectively, and the best model achieved a classification accuracy of 90.37%. These methods are still some ways from practical engineering applications due to the limitation of low detection accuracy; after all, frequent false warnings in the driving process will cause disturbances to the driver and are not conducive to driving safety.

Our study adopts a combination of a manual selection of EEG features and deep learning to give full play to each system’s advantages, aiming to effectively improve detection accuracy. On the one hand, the performance of the EEG decoding algorithm was improved, i.e., a GRU was combined with an EEGNet network to form a GRU-EEGNet network; on the other hand, based on the characteristics of the different power features of the driver’s EEG signals in the θ, α, and β frequency bands when they were driving with distraction and driving with concentration, we first extracted the power features of the θ, α, and β frequency bands of the EEG signals, and then used three GRU-EEGNet models to decode the θ, α, and β band power features. We have three hypotheses: one, that using the method of extracting only the power spectral features of the EEG signal’s θ, α, and β bands is better than the method of extracting the whole features of the five bands of the EEG signal; two, that using the EEGNet model method is better than using an SVM for classification; and three, that the improved GRU-EEGNet has a better classification effect than the EEGNet model. We use a Logitech G29 driving simulator to carry out distracted driving experiments, and the experimental scene and distraction subtasks were constructed by Unity3D. The driver’s EEG signals were synchronously acquired during the experiment, and the acquired EEG signals were preprocessed and processed separately. By extracting the theta, α, and β band power spectral features of the driver’s EEG signals, the SVM and EEGNet models and the GRU-EEGNet method were then used to detect the driving attention state, respectively.

## 2. Materials and Methods

### 2.1. Experiments

#### 2.1.1. Subjects

A total of 30 subjects (15 males and 15 females) were recruited for this study, with an age range of 23 to 31 years (mean = 26.6; SD = 0.88). All subjects were enrolled in master’s or doctoral programs at Wuhan University of Technology and Central China Normal University. They were required to have normal neurological function, normal or corrected-to-normal vision, normal color vision, normal hearing, and a valid driver’s license. The subjects had a range of driving experience, from 1.5 to 9.5 years, with a mean of 4.6 years. Prior to the experiment, they were asked to refrain from consuming alcohol and coffee and to sleep for a minimum of six hours. They were informed of the experiment three days before the commencement of the experiment and were permitted to practice for two hours before the official experiment commenced. This period was designed to familiarize the subjects with the experimental equipment and the experiment until they met the requirements of the experiment. Before the commencement of the experiment, all subjects were provided with written informed consent forms, which indicated that they had volunteered to participate in the experiment and would receive a specified remuneration for their participation.

#### 2.1.2. Experimental Device

A driving simulator and an EEG collector were used to construct the experimental platform (see Figure 1). The driving simulator was composed of a Logitech G29 driving simulator and a 55-inch monitor, and Unity3D was used to design and construct the experimental scene. To simulate the open-road driving scene road conditions as realistically as possible, the test road was a two-way single-lane road with a total length of 5.4 km, separated by a center lane line, with each lane measuring 3.75 m in width. The total number of curves on the road was eight, with radii of 100 m, 50 m, 100 m, 50 m, 150 m, 200 m, 200 m, and 150 m, respectively. Each curve was a linear combination of different turning radii, slopes, and steering. The slope size was divided into two kinds, 4° and 6°, with the curve steering being symmetrical. There was a turn sign in front of each curve. The experimental road was a closed-loop road with no traffic lights and no bifurcation, and there were no pedestrians or other vehicles on the road. The electroencephalogram (EEG) acquisition equipment employed was the ActiCHamp (BRAIN PRODUCTS, Gilching, Germany, https://www.brainproducts.com/solutions/actichamp/, accessed on 27 May 2024), a 64-channel EEG product from Brain Products (BP), which was used to collect real-time EEG data from the driver. During the experimental process, the real-time display provided the driver with information regarding the current number of laps, speed, and the offset of the centerline of the road. This feedback enabled the driver to maintain the speed and control the vehicle to drive along the centerline of the road.

#### 2.1.3. Experimental Paradigm

The experiment required drivers to drive seven laps along the centerline of the lane at a constant speed in a driving simulator, with a maximum speed limit of 90 km/h. Each lap lasted approximately five minutes. Three distraction subtasks were designed: visual distraction, auditory distraction, and cognitive distraction. The three distractor subtasks appeared up to 12 times in each lap, and the types and appearance times of the distractor subtasks were uniformly randomly distributed in each lap. The experiment yielded approximately 12 × 7 = 84 samples per subject; with 30 subjects, there was a total of about 84 × 30 = 2520 samples of the distractor subtasks, of which an average of about 840 samples of each distractor subtask satisfied the requirements of the experimental analysis. It should be noted that the maximum number of 12 distractor subtasks per lap was determined following repeated experiments. Setting too few distractor subtasks will result in an insufficient sample size while setting too many will lead to an interval between two distractor subtasks that is too short, which will affect the experimental effect.

Visual distraction paradigm: The visual distractor subtask was inspired by the HASTE study [30] and another study [31], which simulated a roadside billboard. When the distractor subtask appeared, subjects drove the vehicle while gazing at a visual distractor sub-window that appeared at the top right of the monitor, as shown in Figure 2a. The visual distractor subtask consisted of a single highlighted arrow pattern at the top and a nine-panel grid comprising eight grey arrow patterns pointing in different directions, with an arrowless space at the bottom. These eight arrow patterns were rotated n times, 45° at a time. The arrow pattern highlighted above and the pattern in the nine-panel grid below changed randomly each time it appeared. Subjects were required to select the grey arrow pattern in the 3 × 3 nine-gallery grid that had the same arrow direction as the arrow highlighted above by operating the left and right paddles below the G29 steering wheel. In this paper, a 3 × 3 matrix was coded for the 9-box grid, with the number of times the left paddle of the G29 was dialed down indicating the row values and the number of times the right paddle was dialed down indicating the column values. To select the arrows in the first row and second column shown in Figure 2a to match the upper target, it was necessary to toggle the left paddle once and the right paddle twice. The selected arrow was highlighted to provide the subject with feedback on the selection result. The entire process took 5 s. The system recorded the selection result, the shape of the arrows at each position in the full picture, and the feedback value of the subject’s toggle to determine whether the subject made the correct selection and whether they were distracted. A trial was considered valid as long as the paddle feedback task was completed within the allotted 5 s. However, if the subject did not complete the paddle feedback within the allotted time, the trial was considered invalid and the data from that trial were discarded during later data processing.

Auditory distraction paradigm: The auditory distractor design drew on the n-back memory paradigm. In this experiment, 10 random numbers from 0–9 were played for 1.5 s each. A total of 10 numbers were played, resulting in a total time for one auditory distractor subtask of 15 s. The experiment employed a 2-back paradigm, whereby the subject was required to determine whether the digit played when *x*(*n*) was heard was the same as the digit played at the moment *x*(*n − 2*). If the digits matched, the subject was instructed to press the right paddle to record their judgment. If the digits did not match, the subject was not required to press the paddle and was permitted to continue driving normally, during which time the vehicle was always in motion. The specific process is illustrated in Figure 2b. The system automatically determined whether the subject’s choice was selected correctly based on the subject’s feedback. This experimental paradigm causes both auditory and simple cognitive distractions. However, this approach better fits the reality of distractions caused by drivers talking on the phone or with passengers while driving than a simple auditory distraction like an alarm.

Cognitive distraction paradigm: The cognitive distraction was based on the arrow pattern shown in Figure 2a, combined with a voice announcement prompting the rotation of the arrow highlighted at the top in either a clockwise or counterclockwise direction for a specified number of times (*n* = 1, 2, 3, 4). Each rotation was 45 degrees. This was similar to an announcement prompting the subject to “rotate clockwise 2 times”. Subjects were instructed to select the same arrow as the one highlighted at the top of the nine-panel grid from the eight arrow patterns displayed below and rotate it n times, following the voice announcement prompts. They were then required to use the left and right paddles situated underneath the steering wheel to provide coded feedback. The number of times the left paddle was depressed indicated the row value, while the number of times the right paddle was depressed indicated the column value. The system illuminated the selected arrows by encoding the left and right paddles in the 9-panel grid. The entire process lasted for 10 s. The system recorded the selected arrow shape, the content of the voice cue, and the paddle feedback value at each position in the 9-panel grid to determine whether the subject’s selection was correct and whether the subject was distracted. Should the subject fail to provide the requisite paddle feedback within the allotted 10 s, the trial was deemed invalid, and the data of that trial were discarded during subsequent data processing.

### 2.2. Data Acquisition and Preprocessing

The 64-channel EEG product ActiCHamp acquired EEG signals from the subjects. Before acquisition, the contact impedance between the EEG electrodes and the cortex was calibrated to be less than 5 kΩ. The acquired EEG was filtered using a low-pass filter with a cutoff frequency of 50 Hz and a high-pass filter with a cutoff frequency of 0.5 Hz. This was done to remove DC voltages and high-frequency artifacts. The EEG was then re-referenced to an average of the mastoid electrodes. The portion greater than 100 µV was removed to reduce interference from ocular electricity and muscle activity. The EEG signals were marked according to the mark information recorded during the ActiCHamp acquisition and the event information recorded by Unity3D. The EEG data were then segmented by trial length, with visual distraction for one trial of 5 s, auditory distraction for one trial of 15 s, cognitive distraction for one trial of 10 s, and focused driving for one trial of 5 s. Samples were obtained by intercepting each trial with a time window of one second in length for each of the four driving states separately. This yielded five data samples of one second in length for a visual distraction trial, fifteen data samples of one second in length for an auditory distraction trial, ten data samples of one second in length for a cognitive distraction trial, and five data samples of one second in length for a focused driving trial.

### 2.3. Driving Attention State EEG Power Spectrum Feature Extraction

#### 2.3.1. Power Spectrum Calculation Method

The methods of power spectrum analysis can be broadly classified into two main categories: classical power spectrum calculation methods and modern power spectrum calculation methods. Among these, the classical power spectrum calculation methods include the direct method (periodogram method), the indirect method, and the improved direct method. The modern power spectrum analysis methods encompass calculation methods based on parametric modeling and calculation methods based on nonparametric modeling. In this paper, the improved direct method (Welch’s method) was employed for the calculation of the EEG power spectra.

The Welch method represents an improvement on the Bartlett method. One of the key advantages of the Welch method is that it allows for the data in each segment to be partially overlapped when pairing segments. Additionally, the Welch method permits the data window for each segment to be non-rectangular, with the option of utilizing a Hanning or Hamming window. This mitigates the spectral distortion caused by the rectangular window’s large side flaps, allowing Bartlett’s method to be employed to solve the power spectrum of each segment.

First, we divide the data *x (n)* with a length of *N*, *n* = 0, 1,…, N − 1 into L segments, each with *M* pieces of data. The *i*-th segment is represented as
(1)$$x_{i}\left(n\right)=x\left(n+iM-M\right),\;0\leq n\leq M,1\leq i\leq L$$

The window function *ω(n)* is then added to each data segment to find the periodogram for each segment, and the periodogram for segment *i* is
(2)$$I_{i}\left(\omega \right)=\frac{1}{MU}{\left|\sum_{n=0}^{M-1}x_{i}\left(n\right)\omega \left(n\right)e^{-j\omega n}\right|}^{2},i=1,2,\ldots ,M-1$$

In Equation (2), *U* is called the normalization factor,
(3)$$U=\frac{1}{M}\sum_{n=0}^{M-1}\omega^{2}\left(n\right)$$

Approximating the periodograms of each segment as uncorrelated with each other, the final power spectrum is estimated as
(4)$$P\left(e^{j\omega }\right)=\frac{1}{L}\sum_{i=1}^{L}I_{i}\left(\omega \right)=\frac{1}{MUL}\sum_{i=1}^{L}{\left|\sum_{n=0}^{M-1}x_{i}\left(n\right)\omega \left(n\right)e^{-j\omega n}\right|}^{2}$$

#### 2.3.2. EEG Power Spectrum Analysis of Driving Attention States Based on Welch’s Method

Welch’s method is one of the most frequently employed techniques for calculating EEG power spectral density (PSD). In this paper, the driving attention state EEG data of 30 subjects were selected from the experimental data of 30 subjects for preprocessing separately and then merged to constitute a three-dimensional dataset comprising 100 × 59 × 36,600 elements. The EEG data were resampled at a rate of 100 at 1000 Hz every 1 s, resulting in a total of 100 elements. A total of 64 electrodes were used to collect the electroencephalogram (EEG) signal, with the reference electrode positioned at Fz. Subsequently, the two electrodes most susceptible to eye movement, Fp1 and Fp2, were re-referenced using the electrodes TP9 and TP10. This process resulted in the removal of 59 electrodes, leaving a total of 5 electrodes. A total of 5 electrodes were retained following the removal of data segments deemed to be of poor quality through preprocessing for the driving attention state experiment involving 30 subjects. The above-described Welch’s method was implemented through calculations to obtain the distracted driving and focused driving power spectra, as illustrated in Figure 3. Figure 3A illustrates the power spectral density (PSD) of visual distracted driving and focused driving, Figure 3B depicts the PSDs of auditory distracted driving and focused driving, and Figure 3C presents the PSDs of cognitive distracted driving and focused driving. The figure displays 59 electrodes as horizontal coordinates, and the changes in the power spectra of the 59 electrodes can be observed when driving with distraction and focused driving.

To gain further insight into the power spectra of the driver’s EEG signals in the θ (4–8 Hz), α (8–14 Hz), and β (14–35 Hz) bands at each electrode during distracted driving, this paper calculated the power spectra of the θ, α, and β frequency bands for visual, auditory, and cognitive distraction, respectively, and compared them with those of the θ, α, and β bands for focused driving, as shown in Figure 4. The first column in Figure 4 displays the power spectra of visual distraction and focused driving, the second column displays the power spectra of auditory distraction and focused driving, and the third column displays the power spectra of cognitive distraction and focused driving. The first row in the figure displays the power spectra of the θ frequency band, the second row displays the power spectra of the α frequency band, and the third row displays the power spectra of the β frequency band.

### 2.4. GRU-EEGNet Basic Architecture

#### 2.4.1. EEGNet Network Architecture

This paper employs the EEGNet network, as proposed in the literature [32], for the extraction and classification of driving attention state EEG signals. The network structure is depicted in Figure 5 and Table 1. Following a series of preprocessing steps, including channel localization, filtering, re-referencing, independent component analysis, and others, the experimentally collected EEG signals were fed into the EEGNet network for training.

The EEGNet model parameters are presented in Table 2. The network parameters were set as follows: the batch size was 64, the training period was 300, the adaptive moment estimation (Adam) optimizer was used to train the model with the function value of the cross-entropy loss function as the objective, the learning rate was 0.001, the neuron deactivation parameter of the Dropout layer was set to 0.25, and the model was trained using the 5-fold cross-validation method.

#### 2.4.2. Gated Recurrent Unit

The gated recurrent unit (GRU) is regarded as a variant of a long short-term memory (LSTM) unit. Like the LSTM, it can effectively capture semantic associations between sequences and mitigate the phenomenon of gradient vanishing or explosion. However, its structure and computation are simpler than that of the LSTM. Its core structure can be divided into two parts: the reset gate (red box) and the update gate (blue box), as shown in Figure 6A. The two gating vectors determine which information is ultimately outputted by the gated loop unit. A distinctive feature of these two gating mechanisms is that they maintain information in a long-term sequence, preventing its purging over time or removal due to irrelevance to the prediction.

The GRU unifies the oblivious and input gates of LSTM into a single update gate, also integrates the cell state C and hidden state h of LSTM, and implements additional modifications to render its model in a more streamlined fashion than the standard LSTM model, whose mathematical expression is as follows:(5)$$z_{t}=\sigma \left(W_{z}\cdot \left[h_{t-1},x_{t}\right]\right)$$
(6)$$r_{t}=\sigma \left(W_{r}\cdot \left[h_{t-1},x_{t}\right]\right)$$
(7)$${\tilde{h}}_{t}=\tanh\left(W\cdot \left[r_{t}\times h_{t-1},x_{t}\right]\right)$$
(8)$$h_{t}=\left(1-z_{t}\right)\times h_{t-1}+z_{t}\times {\tilde{h}}_{t}$$

First, the gate values for the update and reset gates, $z_{t}$ and $r_{t}$, respectively, are computed by linear transformation using $x_{t}$ spliced with $h_{t-1}$ and then activated by sigmoid. After that, the reset gate value acts on $h_{t-1}$, representing the control of how much of the information coming from the previous time step can be utilized. Then, the basic RNN computation using this reset $h_{t-1}$ is employed, i.e., it is spliced with $x_{t}$ for a linear variation, which undergoes a tanh activation to get the new ${\tilde{h}}_{t}$, as shown in Equation (7). Finally, the gate value of the update gate will act on the new ${\tilde{h}}_{t}$, while the $1-z_{t}$ gate value will act on $h_{t-1}$, and subsequently, the results of both will be summed up to get the final implicit state output $h_{t}$, as shown in Equation (8). This process implies that the update gate can preserve the previous result, and when the gate value tends toward 1, the output is the new ${\tilde{h}}_{t}$, while when the gate value tends toward 0, the output is $h_{t-1}$ from the previous time step.

The reset gate $r_{t}$ controls the proportion of the information, $h_{t-1}$ from the previous state, that is passed into the candidate state, ${\tilde{h}}_{t}$. The smaller the value of the reset gate, $r_{t}$, the smaller the product with $h_{t-1}$, and the less information is added to the candidate state from $h_{t-1}$. The update gate is used to control how much of the information, $h_{t-1}$ from the previous state, is retained in the new state, $h_{t}$; as $1-z_{t}$ gets larger, more information is retained.

#### 2.4.3. GRU-EEGNet

From the preceding analysis, it can be observed that when driving in a distracted state, the power spectrum of the driver’s electroencephalogram (EEG) signal exhibits a notable divergence from that observed during focused driving. Consequently, this paper proposes a modification to the previous practice of directly transmitting the preprocessed EEG signal to the EEGNet network for training. Instead, we initially derive the power spectra of the EEG signals in the θ, α, and β bands, and subsequently transmit them to the three EEGNets for training, respectively. This enhances the relevance of signal feature extraction and also greatly reduces the number of parameters of the algorithm, thereby speeding up the training process.

Previous studies have demonstrated that EEGNet+LSTM can effectively enhance the performance of EEGNet networks [27]. GRUs and LSTM serve comparable functions in suppressing gradient vanishing or explosion when capturing semantic associations of long sequences. In comparison to LSTM, the utilization of a GRU can achieve comparable results. Furthermore, the number of gates in the GRU unit is smaller, the number of weights and parameters that need to be updated during the training period is smaller, the training speed is faster, and the amount of data needed is less in comparison. This can largely improve the training efficiency. The objective was to enhance the efficiency of the system. To this end, the EEGNet network was combined with the GRU to form the EEGNet + GRU module, which was then used to extract and classify the power spectra of the θ, α, and β frequency bands of the EEG signal associated with the driving attention state. This was achieved by forming three such modules, which collectively constituted the GRU-EEGNet model.

In particular, the preprocessed EEG signals of driving attention states were initially calculated by applying Welch’s method to determine the PSDs of 59 channels within the 0–50 Hz band, which were then converted to dB. Thereafter, the dB values of each channel were normalized to have a mean of zero and a standard deviation of one. Finally, the PSDs of the θ (4–8 Hz) band of 5 × 59, the α (8–13 Hz) band of 6 × 59, and the β (13–35 Hz) band of 23 × 59 were employed as inputs for training purposes. The structure of the GRU-EEGNet network is illustrated in Figure 7.

Three EEGNet+GRU networks were employed to train the PSDs of the three frequency bands separately. The structure of the GRU network is shown in Figure 8. The GRU network was accessed from the middle of the flattened layer and the Dense layer of the EEGNet network, as shown in Figure 8. The parameters of the GRU network were set as follows: the batch size was 16, the training period was 100, and the optimization was performed using the adaptive moment estimation (Adam) optimization apparatus, trained with the function value of the cross-entropy loss function as the optimization objective. The learning rate was 0.001, and the neuron deactivation parameter of the Dropout layer was set to 0.5. The parameters of the EEGNet network were maintained throughout the GRU training process.

## 3. Results

### 3.1. Driving Attention State Detection Based on EEG Power Spectrum Features

Once the power spectral features of the EEG signal representing the driving attention state had been obtained, these were transformed into an N × 59 feature matrix, where 59 is the number of channels and N is the number of trials of the classified event for a particular subject. The classification algorithm was then applied.

#### 3.1.1. Performance Comparison of Common Classification Methods

The most commonly used EEG classification methods are decision trees, discriminant analysis, naive Bayes, support vector machines (SVMs), nearest neighbor classifiers (kNN), and neural networks. To test the classification performance of these six types of classifiers on the power spectrum features of the EEG signal of the driving attention state, the data of six subjects were selected for testing, as shown in Figure 9. The SVM classifier demonstrated optimal performance in both two-class and three-class classification scenarios, as well as in four-class classification scenarios. Therefore, the SVM classifier was selected for the classification and detection of the driving attention state in this paper.

#### 3.1.2. SVM-Based Driving Attention State Detection

The SVM classifier, based on the radial basis kernel function, was selected to classify distracted driving and focused driving based on the extracted power spectrum features of the EEG signal. This classification was conducted on 30 subjects. The previous calculation of the sub-band power spectra of EEG signals revealed that the main differences between distracted driving and focused driving are in the three frequency bands of θ, α, and β. Therefore, in this paper, the power spectral features of these three frequency bands were extracted separately, and then the power spectra of these three frequency bands were combined to form an N × 177 feature matrix, in which 59 × 3 = 177; 59 is the number of channels, 3 is the three frequency bands of θ, α, and β, and N is the number of trials to be classified. Subsequently, the classification was performed with an SVM. The average classification accuracy of the SVM for power spectrum features and three-band combined power spectrum features was calculated for 30 subjects, as shown in Figure 10. A comparison of the results reveals that the power spectrum features constructed using decomposed θ, α, and β bands exhibit higher classification accuracy than those constructed by directly calculating the power spectrum of the EEG signal, particularly for the triple and quadruple classifications, which are more challenging to classify and have lower classification accuracy.

### 3.2. EEGNet and GRU-EEGNet Classification Results

The EEGNet and GRU-EEGNet network models were employed to identify driving attention states. The preprocessed driving attention state EEG signals were utilized to train the EEGNet network model, which was subsequently employed to classify the driving attention state. In contrast to EEGNet, the training and testing datasets for the GRU-EEGNet network comprised the power spectral densities of the driving attention state EEG signals in θ, α, and β bands.

EEGNet and GRU-EEGNet were employed to categorize the four driving attention states of 30 subjects, namely, focused, visual, auditory, and cognitive distraction, respectively. The performances of the three methods, namely the three-band combined feature PSD_SVM method, EEGNet, and GRU-EEGNet, were then compared. The average detection accuracies of the three methods for the 30 subjects’ driving attentional states are presented in Figure 11A. The classification performance of the GRU-EEGNet and EEGNet methods is superior to that of the PSD_SVM method, regardless of the number of classifications. For instance, the classification accuracies of the PSD_SVM, EEGNet, and GRU-EEGNet methods for the two-class classification of visual distraction and focused driving are 76.67%, 85.65%, and 91.45%, respectively. It is evident that regardless of the employed method, the accuracy of the two-class classification is the highest, while that of the four-class classification is the lowest. Furthermore, the classification accuracy tends to decrease with an increase in the number of classification categories.

Figure 11B illustrates the classification performance of the EEGNet and GRU-EEGNet models for 30 subjects’ driving attention states. The models exhibited a notable enhancement in the performance of GRU-EEGNet over EEGNet in the second, third, and fourth classifications. In the figure, E(2) denotes EEGNet’s two-class classification, G-E(3) denotes GRU-EEGNet’s three-class classification, and so on for the others. Table 3 presents the four-class classification performance of the GRU-EEGNet model for 10 subjects. For instance, in the case of the tri-classification of visual, auditory, and focused driving, the classification accuracies of EEGNet and GRU-EEGNet were 79.71% and 88.75%, respectively. The classification accuracy of GRU-EEGNet was 9.04% higher than that of EEGNet.

### 3.3. Online Detection System of Driving Attention State Based on EEG

#### 3.3.1. System Structure and Function

To assess the efficacy of the algorithm, this paper presents an EEG-based driving attention state detection system for online experimental testing. The system is comprised of four principal components: human–computer interaction, a data acquisition module, a data processing module, and a display module, as illustrated in Figure 12.

The human–computer interaction interface is the primary means of controlling the model selection and initiating and terminating the driving attention state detection system, and serves as the conduit for user–system interaction. Users can manipulate the experimental process through the use of virtual buttons.

The data acquisition module is responsible for acquiring the EEG signal of the subject in real time and transmitting it to the data processing module.

The data processing module is responsible for handling the EEG signals sent from the receiving data acquisition module. It sequentially preprocesses the signals with a range of techniques, including channel selection, re-referencing, filtering, downsampling, and so forth. The preprocessed data is then fed into the trained EEGNet and GRU-EEGNet models for prediction, with the prediction results subsequently sent to the display module for display.

The display module is designed to provide real-time feedback on the driver’s driving state. These states include focused driving, visual distraction, auditory distraction, and cognitive distraction.

#### 3.3.2. System Module Realization

(1)The human–computer interface was designed by QT Designer. The left side of the interface is for model selection, while the right side is for experimental control and test result display. This is illustrated in Figure 13. The model selection is realized by QFileDialog control, whereby the path of the model can be selected through the file selection dialog box. The experimental control section is implemented using a push-button control, which modifies the trigger signal via the button and calls the associated slot function to execute the corresponding function. The test result display function is a LineEdit text box, which transforms the predicted value transmitted from the data processing module into the corresponding driver state and displays it in the text box.(2)The EEG data acquisition module facilitates the real-time transmission of EEG signals to a computer terminal via the remote data transmission RDA (Remote Data Access) interface provided by Pycorder software (https://brainlatam.com/manufacturers/brain-products/pycorder-127, accessed on 8 August 2022). This is achieved through the transmission of the EEG signal based on the TCP, with the signal acquisition module triggering a transmit signal to send the data to the data processing module after acquiring the data for the specified length of time (here, 1 s).(3)The data processing module is described here. The EEG data captured by ActiCHamp in real time is of a list type without channel information. The EEG data are converted to a RawArray type using Python’s MNE toolkit, and then preprocessed sequentially with channel selection, re-referencing, filtering, and downsampling. The online preprocessing is consistent with the offline preprocessing, including the selection of the same channels (59 channels or 10 channels), re-reference electrodes (TP9, TP10), filtering parameters (0.1–45 Hz), and downsampling rate (100 Hz). The preprocessed data are directly fed into the trained GRU-EEGNet model for feature extraction and classification.(4)The display module is responsible for converting the predicted values generated by the data processing module into the corresponding driver states and displaying them in a text box.

#### 3.3.3. Online Experiment and Effect

A total of five subjects were selected for an online experiment on driving attention states. As in the offline experiment, subjects drove the vehicle along the centerline of the road for three laps on simulated driving equipment, with each lap lasting approximately five minutes. Visual, auditory, and cognitive distractor subtasks appeared randomly during driving, with a maximum of 12 distractor subtasks per lap. The durations of the visual, auditory, and cognitive distractors were designed to be 5 s, 15 s, and 10 s, respectively. The system detected the data at 1 s intervals, resulting in the occurrences of visual, auditory, and cognitive distractions being counted 5 times, 15 times, and 10 times, respectively. The time and number of occurrences of each distractor task during the entire experiment were recorded, and the detection output of the model per second during the experiment was recorded in real time. Categorical statistics were performed at the end of the experiment, as shown in Table 4. The detection accuracies of the four driving attentional states were 73.94% and 77.30% for EEGNet and GRU-EEGNet, respectively, when tested offline. The detection accuracies of the four driving attentional states were 66.25% and 71.71% when tested online by five subjects, representing decreases of 7.69% and 5.59%, respectively, with respect to the offline test. Similar to the offline test, the detection accuracy of GRU-EEGNet was 5.46% higher than that of EEGNet during the online test.

## 4. Discussions

Feature extraction and classification are two key techniques for EEG decoding. Feature extraction of EEG signals is a prerequisite for their classification. The main traditional machine learning feature extraction methods for EEG signals are time–domain, frequency–domain, time–frequency–domain, and statistical feature-based methods. One or more combinations of features of the signal can be selected, e.g., using singular value decomposition (SVD) [26] and wavelet analysis [33] to extract driver EEG features for driving state detection. In our study, the theta, alpha, and beta activities of driver EEG signals during distracted versus focused driving were significantly different; this finding is based on the fact that the activities of these three frequency bands differed from each other in different brain regions for different types of distracted driving. Therefore, extracting the power spectrum features of the theta, α, and β frequency bands of the EEG signals enables the classification of different driving attention states. We hypothesized that this method of extracting power spectral features of only three frequency bands, θ, α, and β, would be better compared to the method of extracting power spectral features of the whole frequency band of the EEG signal. Because this method removes the useless features and strengthens the feature differences to make them more separable, the experimental results prove our hypothesis.

The traditional machine learning classification methods for distracted driving EEG signals are the support vector machine (SVM) [34], Bayes Classifier [35], k-Nearest Neighbor (kNN) [36], decision tree (DT) [37], and Artificial Neural Network (ANN) [38], among others. To find out the classification methods that can classify the extracted EEG power spectrum features well, we compared six methods, namely the DT, discriminant analysis, naive Bayes, SVM, kNN, and ANN; the classification performance of these six methods was found to be the best for the SVM.

Aiming at the problem that feature extraction of traditional machine learning EEG decoding methods requires manual experience or a priori knowledge, and the decoding accuracy is limited by the low signal-to-noise ratio and spatial resolution of the EEG signals, some researchers have used a pure data approach to implement an end-to-end deep learning method to decode driving distraction EEG signals. For example, a CNN was used to implement driving load detection [39], an RNN was used to implement cognitive distraction detection [40,41], and LSTM [42] and a GRU [28] were used to detect distraction. However, these methods require a large amount of data to train the model and suffer from overfitting and poor inter-subject robustness. One solution idea is to fuse EEGs with other features to improve the accuracy of driving distraction detection, such as using sliding window multiscale entropy (MSE) and a bi-directional long and short-term memory (BiLSTM) network. Sliding window MSE was used to extract EEG features to determine the location of a distraction, and then statistical analysis of vehicle behavioral data was performed; finally, a BiLSTM network was used to combine MSE and other conventional features to detect driver distraction, and better detection results were achieved [43]. A multi-information fusion of electroencephalograms (EEGs), electrocardiograms (ECGs), electromyograms (EMGs), and electrooculograms (EOGs), along with behavioral information (PERCLOS70-80-90%, mouth aspect ratio (MAR), eye aspect ratio (EAR), blinking frequency (BF), and head-to-head ratio (HT-R)), was used to detect distraction, by first using recurrent neural network and long short-term memory (RNN-LSTM) models to extract physiological features. After fusing these features, classification was performed using a deep residual neural network (DRNN), and a detection accuracy of 96.5% was achieved [44]. However, the multi-sensor fusion method also suffers from the problem of data synchronization and is not easy to use in practical engineering applications. Another solution idea is to combine two or more network models and give full play to the advantages of each network to improve the detection accuracy. For example, CNN and GRU models were combined to extract the spatio-temporal features of EEG signals, taking advantage of GRUs’ expertise in solving sequential problems; the accuracy of two-class classification (distracted and non-distracted) ranged from 82% to 100%, and the accuracy of three-class classification (mobile phone operation distracted, two-back distracted, and non-distracted) ranged from 76% to 98% [28]. By combining LSTM with EEGNet for triple classification of visual, auditory, and focused driving, researchers achieved an average accuracy of 71.1% [29]. The third solution idea is to dig deeper into the performance of deep learning network models by optimizing the training method [45].

Our study innovatively combines the provision of EEG features based on manual experience with an improved decoding algorithm. On the one hand, based on the results of existing studies, the initial selection of EEG features is made, which is different from the conventional practice of extracting the full-band features of the EEG. This is replaced by the extraction of the power spectrum features of the θ, α, and β bands, which are the most distinctive and separable features. On the other hand, the combination of a GRU and EEGNet is used to form three GRU-EEGNet networks to further feature extraction and classification of the extracted power spectrum features in the θ, α, and β bands, respectively. The experimental results prove that the method achieves better results compared with existing similar methods.

Although the method proposed in this paper achieves better results, there are some limitations in this study. (1) The experiments in this study are based on a driving simulator, with the simulation completed in an ideal driving environment with no traffic flow. Although the online effect is verified, the validity of the experimental results has not been tested by real vehicle experiments in an actual open-road environment. (2) In order to be more in line with real driving road conditions, we used a road with many curves and slopes, which caused the drivers to frequently adjust the steering wheel and step on the accelerator and brake pedal; the frequent body movements interfered with the EEG signals, and we were not able to filter out these artifacts completely, even though we used a variety of methods. In the next step of the study, we plan to (1) use the EEG acquisition equipment used in this study and have subjects drive a real car to conduct experiments in open-road scenarios to test the effectiveness of the method; (2) continue to improve the EEG decoding algorithm to improve detection accuracy and robustness; (3) downsize the EEG electrode channels and use dry electrodes to make the EEG acquisition wireless, portable, and wearable, as wearing a hat is convenient, to facilitate engineering applications and promotion; 4) combine the experiment with other features, such as the fusion of multi-sensor information such as that from cameras, EMGs, and in-vehicle driving performance sensors, to improve the detection accuracy and robustness of driving distraction detection by exploiting the strengths and avoiding the weaknesses of the current method.

## 5. Conclusions

This paper is based on the observation that there are clear differences in the power spectra of a driver’s EEG signal’s θ, α, and β frequency bands during distracted driving and focused driving. We have chosen to extract the PSD features of the EEG signal’s θ, α, and β frequency bands during different driving attention states and use an SVM for classification to achieve the detection of driving attention states. To enhance the precision of the detection process, a novel approach, the GRU-EEGNet method, has been developed. This method leverages the strengths of both the EEGNet network and the gated recurrent unit (GRU) network to enhance the accuracy of driving attention state detection. Its efficacy has been validated through rigorous online experiments. It is found that the adopted approach of extracting only theta, alpha, and beta band power spectrum features of the EEG signals is more effective in classification than the previous approach of extracting full-band power spectrum features. The driving attention state detection accuracy of the proposed GRU-EEGNet model is improved by 6.3% and 12.8% over the EEGNet model and PSD_SVM method, respectively. This study lays the foundation for the next engineering application of EEG-based driving attention state detection, which will not only enrich the functioning of assisted driving systems but also effectively improve driving safety performance.

## Figures and Tables

**Figure 1 sensors-24-05086-f001:**
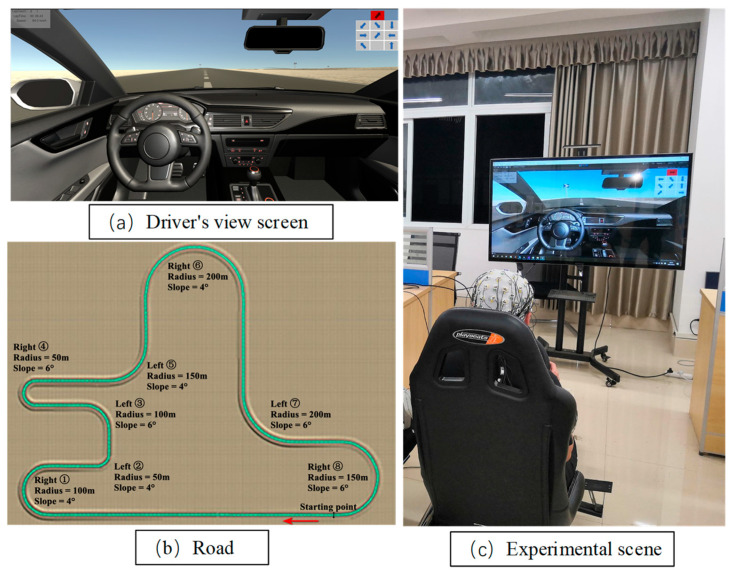
Simulated driving experimental platform. (**a**) Simulated driving display with current lap and vehicle speed parameters displayed in the upper left corner and visual distraction subtasks displayed in the upper right corner. (**b**) Experimental road. (**c**) The experimental scene with a subject wearing an EEG cap on their head.

**Figure 2 sensors-24-05086-f002:**
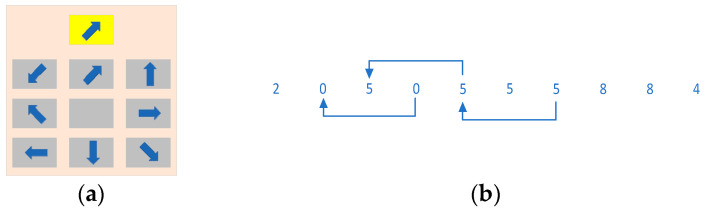
Visual and auditory distraction paradigm. (**a**) Visual distraction paradigm. (**b**) Auditory distraction paradigm.

**Figure 3 sensors-24-05086-f003:**
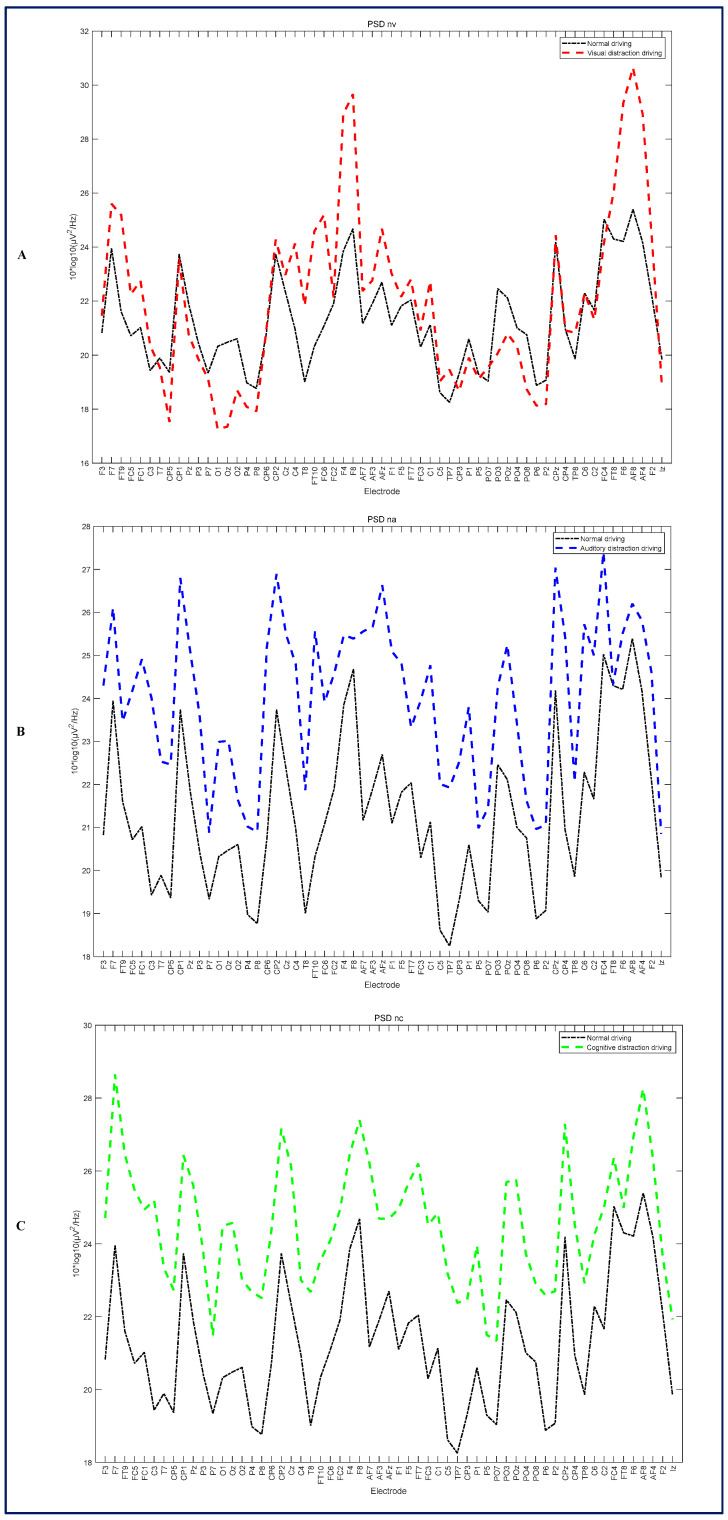
Distracted versus focused driving power spectra by channel. (**A**) Visual distracted versus focused driving power spectra; (**B**) auditory distracted versus focused driving power spectra; (**C**) cognitive distracted versus focused driving power spectra.

**Figure 4 sensors-24-05086-f004:**
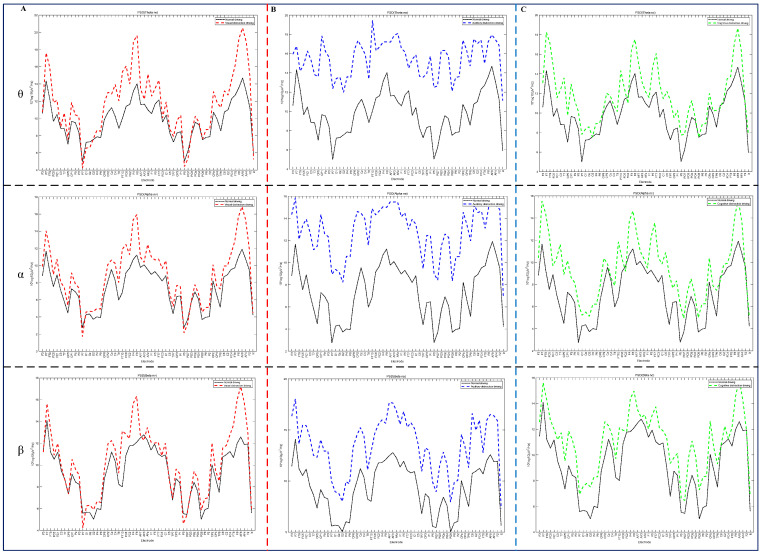
Power spectra of each channel in the θ, α, and β bands of the driving attention state. (**A**) visual distraction vs. focused driving; (**B**) auditory distraction vs. focused driving; (**C**) cognitive distraction vs. focused driving.

**Figure 5 sensors-24-05086-f005:**
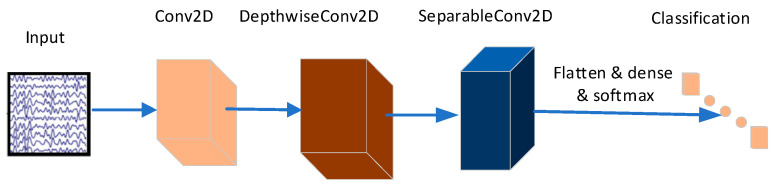
Schematic diagram of EEGNet network structure.

**Figure 6 sensors-24-05086-f006:**
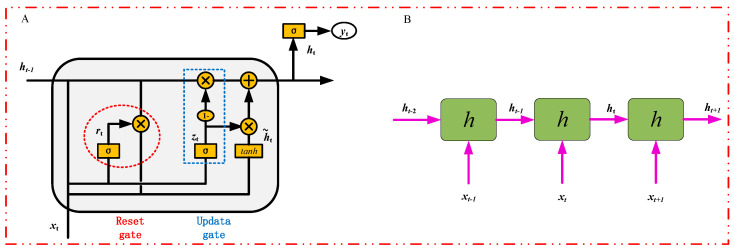
Schematic diagram of the internal structure of the GRU. (**A**) schematic diagram of a unit GRU structure; (**B**) schematic diagram of three GRU units connected in stages.

**Figure 7 sensors-24-05086-f007:**
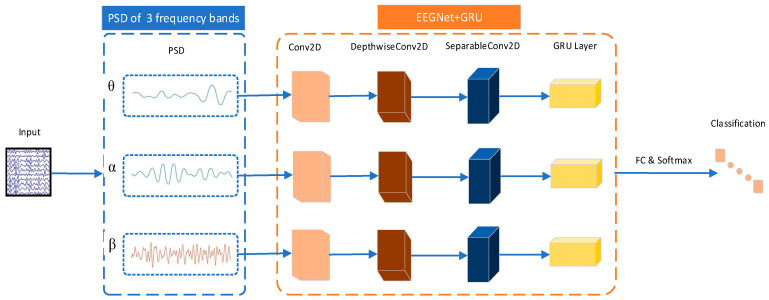
Schematic of EEG decoding for GRU-EEGNet model.

**Figure 8 sensors-24-05086-f008:**
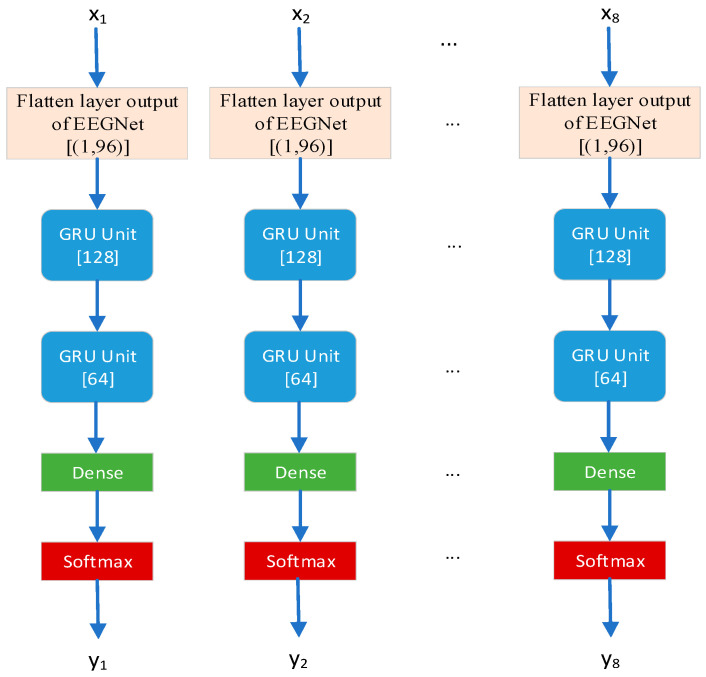
GRU network structure and parameters.

**Figure 9 sensors-24-05086-f009:**
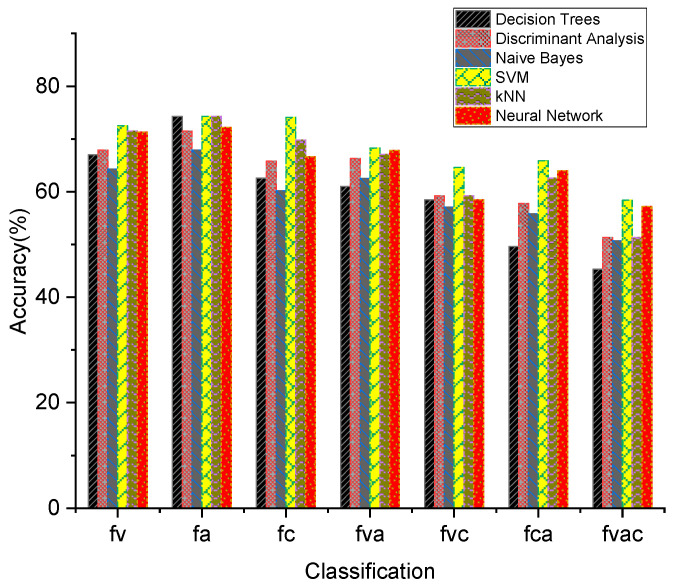
Classification performance of six commonly used EEG classification methods (f: focused; v: visual; a: auditory; c: cognitive; fv: focused vs. visually distracted driving classification; fa: focused vs. auditory distracted driving classification; fc: focused vs. cognitive distracted driving classification; fva: focused, visually distracted vs. auditorily distracted driving triple classification; fvc: focused, visually distracted vs. cognitive distracted driving triple classification; fca: focused, cognitive distracted vs. auditorily distracted driving triple classification; fvac: focused, visually distracted, auditorily distracted vs. cognitively distracted driving quadruple classification).

**Figure 10 sensors-24-05086-f010:**
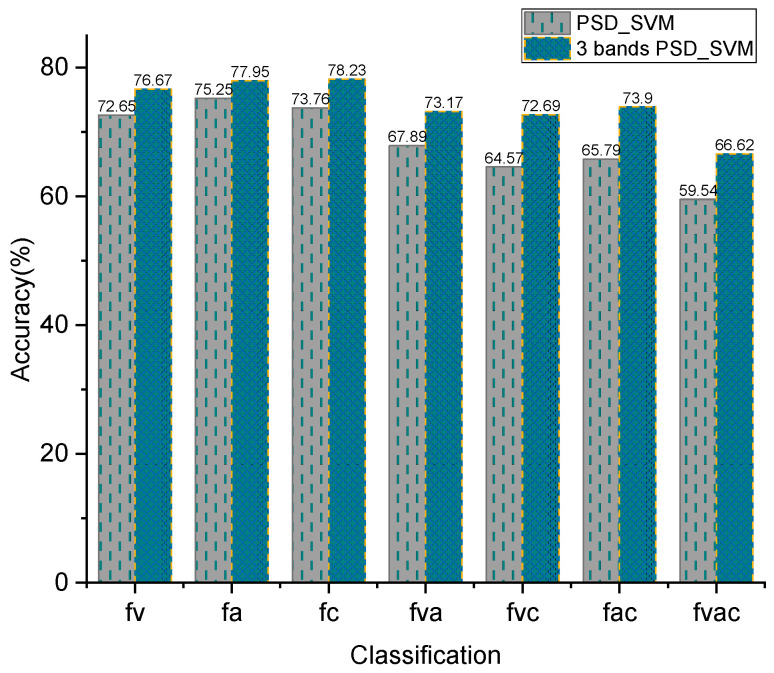
Average classification accuracy of full-band and three-band feature extraction methods.

**Figure 11 sensors-24-05086-f011:**
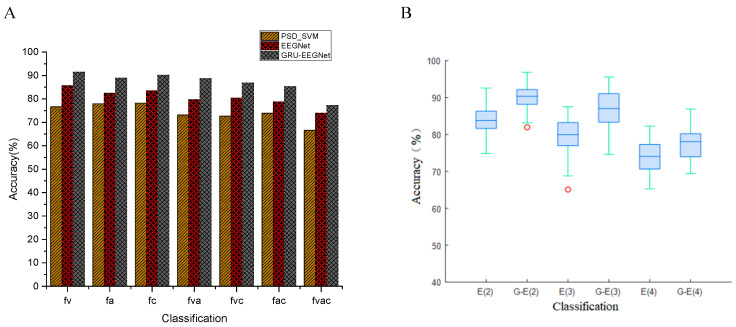
EEGNet and GRU-EEGNet classification performance. (**A**) Average classification accuracy of PSD_SVM, EEGNet, and GRU-EEGNet; (**B**) performance of EEGNet and GRU-EEGNet (The horizontal line in the box represents the median, and the red circle represents the outlier). (E(2): EEGNet 2 classification; G-E(3): GRU-EEGNet 3 classification).

**Figure 12 sensors-24-05086-f012:**
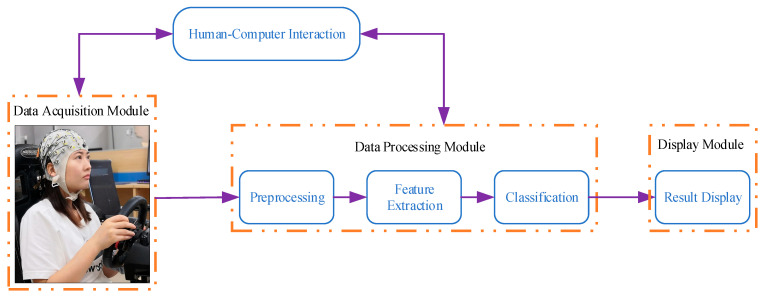
Schematic structure of driving attention state detection system.

**Figure 13 sensors-24-05086-f013:**
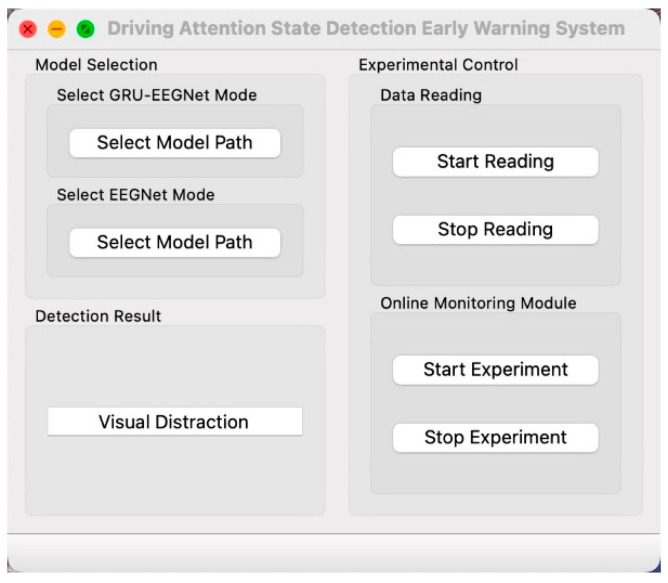
Driving attention state detection system interface.

**Table 1 sensors-24-05086-t001:** EEGNet architecture.

Block	Layer	# Filters	Size	Output	Activation
1	Input			(C, T)	
	Reshape			(1, C, T)	
	Conv2D	F_1_	(1, 64)	(F_1_, C, T)	Linear
	BatchNorm			(F_1_, C, T)	
	DepthwiseConv2D	D * F_1_	(C, 1)	(D * F_1_, 1, T)	Linear
	BatchNorm			(D * F_1_, 1, T)	
	Activation			(D * F_1_, 1, T)	ELU
	AveragePool2D		(1, 4)	(D * F_1_, 1, T//4)	
	Dropout *			(D * F_1_, 1, T//4)	
2	SeparableConv2D	F_2_	(1, 16)	(F_2_, 1, T//4)	Linear
	BatchNorm			(F_2_, 1, T//4)	
	Activation			(F_2_, 1, T//4)	ELU
	AveragePool2D		(1, 8)	(F_2_,1,T//32)	
	Dropout *			(F_2_, 1, T//32)	
	Flatten			(F_2_ * (T//32))	
Classifier	Dense	N * (F_2_ * T//32)		N	Softmax

Table 1: EEGNet architecture, where C is the number of channels, T is the number of time points, F1 is the number of temporal filters, D is the depth multiplier (number of spatial filters), F2 is the number of point filters, and N is the number of categories. For the Dropout layer, we use *p* = 0.5 for within-subject classification and *p* = 0.25 for cross-subject classification. “*” indicating the matrix dimension.

**Table 2 sensors-24-05086-t002:** EEGNet model parameters.

Model Structure	Convolution Kernel Size	Number of Convolution Kernels	Step Size
Conv2D	(1,50)	16	1 × 1
DepthwiseConv2D	(59,1)	32	1 × 1
AveragePool2D	(1,4)	-	1 × 4
SeparableConv2D	(1,16)	32	1 × 1
AveragePool2D	(1,8)	-	1 × 8

**Table 3 sensors-24-05086-t003:** Performance metrics for driving attention state classification in GRU-EEGNet network.

		Sub1	Sub2	Sub3	Sub4	Sub5	Sub6	Sub7	Sub8	Sub9	Sub10	Avg.
Acc(%)		69.42	76.57	86.93	78.36	70.70	78.65	71.26	80.32	79.53	85.19	77.69
K value		0.664	0.728	0.842	0.741	0.653	0.746	0.672	0.784	0.768	0.827	0.743
F1 score		0.695	0.766	0.870	0.784	0.708	0.787	0.713	0.803	0.796	0.853	0.778
Precision	V	0.732	0.835	0.913	0.836	0.744	0.833	0.766	0.736	0.861	0.914	0.817
	A	0.638	0.714	0.823	0.766	0.668	0.802	0.674	0.813	0.739	0.798	0.743
	C	0.714	0.726	0.854	0.744	0.734	0.749	0.744	0.871	0.776	0.828	0.774
	F	0.695	0.788	0.888	0.790	0.685	0.764	0.669	0.794	0.814	0.868	0.775
	Avg.	0.695	0.766	0.869	0.784	0.708	0.787	0.713	0.803	0.797	0.852	0.777
Recall	V	0.746	0.774	0.846	0.766	0.727	0.774	0.768	0.822	0.773	0.887	0.788
	A	0.689	0.809	0.878	0.828	0.709	0.794	0.715	0.785	0.852	0.844	0.790
	C	0.641	0.756	0.853	0.784	0.676	0.825	0.662	0.754	0.798	0.896	0.765
	F	0.709	0.728	0.901	0.760	0.719	0.759	0.710	0.853	0.758	0.786	0.768
	Avg.	0.696	0.767	0.870	0.785	0.707	0.788	0.714	0.803	0.796	0.853	0.778

**Table 4 sensors-24-05086-t004:** Performance of online detection of driving attention states before and after dimensionality reduction of GRU-EEGNet model.

Model	Subject	Visual Distraction		Auditory Distraction		Cognitive Distraction		Focused Driving		Predictive Accuracy (%)
		T	C	T	C	T	C	T	C	
GRU-EEGNet	Sub1	65	52	150	72	100	69	608	486	73.56%
	Sub2	55	44	180	86	130	76	576	435	68.12%
	Sub3	60	47	180	91	110	81	648	486	70.64%
	Sub4	65	54	165	71	120	83	582	479	73.71%
	Sub5	60	51	150	69	130	79	617	495	72.52%
EEGNet	Sub1	60	49	165	76	130	79	584	425	66.99%
	Sub2	65	51	150	76	120	75	612	413	64.94%
	Sub3	50	41	195	94	110	73	638	456	66.87%
	Sub4	55	46	180	89	120	82	596	421	67.09%
	Sub5	60	47	180	85	110	71	629	437	65.37%

Note: T: total number; C: number of correct results.

## Data Availability

The data presented in this study are available on request from the corresponding author.

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
