# Peer review of "Driving Attention State Detection Based on GRU-EEGNet"

_sensors, 2024, doi:10.3390/s24165086_

Round 1
Reviewer 1 Report
Comments and Suggestions for Authors
The paper attempts to address two gaps in research literature: namely, the use of EEG for monitoring driving and a comparison of analytical methods (SVM, EEGNet, and GRU-EEGNet) put forth by the authors. The authors look at differences in θ, α, and β band power spectra in EEG, in terms of whether individuals will be distracted or focused during driving. The analysis using θ, α, and β band power spectra of EEG signals, with the SVM, EEGNet, and GRU-EEGNet models is a strong analysis and offers in and of itself a contribution to EEG research. References referring to the analytical methods used in the research are similarly appropriate. The manuscript is of high technical and methodological quality. However, the stated goal of utilizing eeg for practical application for driving attentiveness seems not well explained in its implementation in real life. While the authors put consideration into the three subtasks that they selected, namely visual distraction, auditory distraction and cognitive distraction, the use of driving simulation in this case is not well explained as to its practical value. While simulation may be an excellent tool for understanding the impact of many phenomenon, the technology tested (in this case, EEG), should use a version of that technology that could also be implemented in real life, and an explanation of how that implementation could be done in real life should be included. In this case, little attempt is made to explain how EEG would be utilized in reality while a person is driving a real car in a real environment similar to that represented in the simulation. Moreover, the EEG unit used during the simulation should be one that would work easily for average drivers - the masters or doctoral students used in the study are not necessarily representative of having the same skill set with handling EEG devices as average drivers, though that is not my main concern. ActiCHamp, the 64-channel EEG product from Brain Products (BP), which was used to collect real-time EEG data from the drivers, is far from being a consumer device, and it is difficult to even imagine a PhD level researcher driving an actual car while wearing such a unit.
The decision to use simulation, while understandably convenient, raises the question, for EEG in particular, of is the actual driver to wear an eeg headset, or what kind of wearable or remote sensing is needed? Minor corrections are needed to address the practicality issue, in my opinion, which should contain some additional references, noting the use of simulation with EEG to practical purpose, and how that extends to driving in implementation. If this is explainable by the authors, in the article discussion and perhaps other portions of the paper, then the paper would offer a very significant contribution, in laying out a practical method, using new analytical methods, of utilizing EEG for driving monitoring. In doing so, citations of how such an application has worked previously in research or practice is suggested, indicating in detail how such prior research either directly demonstrates that practical value with real-life driving comparison, or alternatively, at least a theoretical analysis of how simulation research with EEG, and EEG in particular, could be altered in practicality or engineering methodology, to allow for practical driver use during real-life driving.
Author Response
- Comment: The paper attempts to address two gaps in research literature: namely, the use of EEG for monitoring driving and a comparison of analytical methods (SVM, EEGNet, and GRU-EEGNet) put forth by the authors. The authors look at differences in θ, α, and β band power spectra in EEG, in terms of whether individuals will be distracted or focused during driving. The analysis using θ, α, and β band power spectra of EEG signals, with the SVM, EEGNet, and GRU-EEGNet models is a strong analysis and offers in and of itself a contribution to EEG research. References referring to the analytical methods used in the research are similarly appropriate. The manuscript is of high technical and methodological quality. However, the stated goal of utilizing eeg for practical application for driving attentiveness seems not well explained in its implementation in real life. While the authors put consideration into the three subtasks that they selected, namely visual distraction, auditory distraction and cognitive distraction, the use of driving simulation in this case is not well explained as to its practical value.While simulation may be an excellent tool for understanding the impact of many phenomenon, the technology tested (in this case, EEG), should use a version of that technology that could also be implemented in real life, and an explanation of how that implementation could be done in real life should be included. In this case, little attempt is made to explain how EEG would be utilized in reality while a person is driving a real car in a real environment similar to that represented in the simulation. Moreover, the EEG unit used during the simulation should be one that would work easily for average drivers - the masters or doctoral students used in the study are not necessarily representative of having the same skill set with handling EEG devices as average drivers, though that is not my main concern. ActiCHamp, the 64-channel EEG product from Brain Products (BP), which was used to collect real-time EEG data from the drivers, is far from being a consumer device, and it is difficult to even imagine a PhD level researcher driving an actual car while wearing such a unit.
Response: Special thanks to the reviewer for the suggestions. The introduction section of the original paper did not sufficiently analyze the limitations of previous studies and the importance and necessity of this study. We made significant revisions based on the reviewers' suggestions. The revised introduction consists of 5 paragraphs. Paragraph 1 introduces the importance of driving distractions on traffic safety. Paragraph 2 analyses the advantages and disadvantages of five common driving distraction detection methods, and draws out the advantages of using the EEG method to detect the driving attention state, highlighting the value and significance of this study. Paragraph 3 introduces the feature differences of EEG signals in different driving states, which is the theoretical basis for our manual selection of EEG features. Paragraph 4 introduces the existing research results of driving attention detection based on EEG signals, and the problems that exist. Paragraph 5 presents the research methodology, proposed hypotheses, and experimental methods of this study. In the discussion section of the paper, future research directions are presented, describing how EEG can be used in reality for driving attention state detection. This study was conducted during the summer holidays in China, during which only Masters and PhDs are still in school, so all subjects recruited for this study were Masters and PhDs.
The act of driving is a complex task that requires the coordinated use of a range of cognitive skills, including planning, memory, motor control, and visual abilities. These abilities vary considerably from one individual to another and depend on cognitive skills and attention levels [1]. Distracted attention reduces the driver's level of perception and decision-making ability, negatively affects vehicle control, and is a significant factor in a large number of accidents[2]. The National Highway Traffic Safety Administration (NHTSA) reports that a total of 39,221 fatal crashes occurred in the United States in FY2022, involving 60,048 drivers and resulting in 42,514 deaths. Of these, 3,047 crashes were related to driver distraction, accounting for 8% of all fatalities[3]. Several studies have demonstrated that distraction while driving impairs driving performance[4-13]. If driving distraction can be detected in a timely and effective manner and drivers can be intervened, the degree of injury or even the accident can be reduced or even avoided. Therefore, the research on driving distraction detection methods is of great significance to improve driving safety. (1. Introduction, paragraph 1).
There are five common methods for detecting driving distractions: performance-based, driver body posture, driver physiological characteristics, subjective report, and hybrid detection[14]. However, all of these methods have certain limitations; it is difficult to distinguish between types of distraction in driving performance-based methods, the use of a camera to analyze the driver's body posture to detect distraction is susceptible to light (e.g., daytime and nighttime), the subjective report is subject to subjectivity, and the mixed measurements suffer from the problem of difficulty in synchronizing the data from multiple data sources. EEG, on the other hand, is the result of a series of drivers' mental activities and is characterized by objective accuracy, high temporal resolution, and the ability to detect different types of distraction. In particular, compared with other methods that can only detect distraction when it has already occurred or after it has occurred, EEG technology can be detected when the distraction has just occurred. This is because physiological changes begin when the driver is in the initial stages of distraction. Therefore, driving distraction detection using EEG technology is an effective means to improve driving safety. (1. Introduction, paragraph 2).
Differences in the characteristics of EEG signals in drivers' different driving attention states were found and used to detect distraction. EEG signals theta (4-8 Hz), alpha (8-13 Hz) and beta (13-35 Hz) activities during driving distraction are significantly different from those during focused driving[15]. And this difference varies for different types of distraction. The EEG signal alpha spindle rate was significantly higher during auditory distraction than during focused driving without a subtask[16]. Right parietal alpha and beta activity were negatively correlated with the degree of visual distraction[17]. Alpha and theta band power were positively correlated with cognitive load[18]. This difference has also been demonstrated using neuroimaging to study driving attention states and brain region activation, which found significant activation in the right frontal cortical regions of drivers during distraction[19, 20]. (1. Introduction, paragraph 3).
EEG technology has been widely used for the detection of distracted driving, such as visual distraction[17, 21], auditory distraction[22, 23], and cognitive distraction[24-27], due to its ease of portability, insensitivity to head motion, and high temporal resolution. However, EEG also has the characteristic of a low signal-to-noise ratio due to the signal being susceptible to environmental noise, as well as physiological activity and motion artifacts, which are difficult to eliminate, resulting in poor accuracy of distraction detection. Therefore, current research focuses on EEG decoding algorithms to improve the accuracy of distraction detection. Some studies used traditional machine learning algorithms for EEG decoding, such as using singular value decomposition (SVD) to extract the features of each frequency band of the EEG signal, and then using SVM to classify them to achieve the detection of the driver's cognitive state, which achieved a detection accuracy of 96.78%[28]. Other studies used deep learning methods for decoding, such as Yan et al [29] used EEGNet combined with LSTM to detect visual and auditory distraction, and obtained an average accuracy of 71.1%. Li et al[30]used CNN combined with GRU (Gate Recurrent Unit) for dichotomous classification (distracted vs. non-distracted), and the accuracy between 82%-100% and triple classification (mobile phone operation distraction, 2-back distraction and non-distraction) between 76%-98%.Wang et al [31]performed driver cognitive load detection by extracting the EEG signal θ, α, β, and γ band power spectra by using neural network and SVM, respectively, and the best model achieved a classification accuracy of 90.37%. These methods are still some way from practical engineering applications due to the limitation of low detection accuracy, after all, frequent false warnings in the driving process will cause disturbances to the driver and are not conducive to driving safety. (1. Introduction, paragraph 4).
Our study adopts a combination of manual selection of EEG features and deep learning to give full play to each other's advantages, aiming to effectively improve the detection accuracy. On the one hand, the performance of the EEG decoding algorithm is improved, i.e., GRU is combined with EEGNet to form a GRU-EEGNet network; on the other hand, based on the characteristics of different power of the driver's EEG signals in the θ, α, and β frequency bands when driving with distraction and driving with concentration, we firstly extracted the power features of the θ, α, and β frequency bands of the EEG signals, and then used three GRU-EEGNet models to decode the θ, α and β band power features are then decoded using three GRU-EEGNet models. We have three hypotheses: one using the method of extracting only the power spectral features of the EEG signal θ, α and β bands is better than the method of extracting the whole features of the five bands of the EEG signal, two using the EEGNet model method is better than the SVM for classification, and three the improved GRU-EEGNet has a better classification effect than the EEGNet model. We use a Logitech G29 driving simulator to carry out distracted driving experiments, the experimental scene and distraction sub-tasks are constructed by Unity3D. The driver's EEG signals were synchronously acquired during the experiment, and the acquired EEG signals were pre-processed and processed separately. By extracting the theta, α and β band power spectral features of the driver's EEG signals, the SVM and EEGNet models and the GRU-EEGNet method were then used to detect the driving attention state, respectively. (1. Introduction, paragraph 5).
Although the method proposed in this paper achieves better results, there are some limitations in this study.1) The experiments in this study are based on a driving simulator, which is completed in an ideal driving environment with no traffic flow. Although the online effect is verified, the validity of the experimental results has not been tested by real vehicle experiments in an actual open-road environment. 2) For example, to be more in line with real driving road conditions, we used a road with many curves and slopes, which would cause the driver to frequently adjust the steering wheel and step on the accelerator and brake pedal, and the frequent body movements would interfere with the EEG signals, and we were not able to filter out these artifacts completely, even though we used a variety of methods. In the next step of the study, 1) use the EEG acquisition equipment in this study and drive a real car to conduct experiments in open road scenarios to test the effectiveness of the method; 2) continue to improve the EEG decoding algorithm to improve detection accuracy and robustness; 3) downsize the EEG electrode channels and use dry electrodes to make the EEG acquisition wireless, portable, and wearable, as wearing a hat is convenient, to facilitate engineering applications and promotion; 4) Combine with other features, such as fusion of multi-sensor information such as camera, EMG, and in-vehicle driving performance sensors, to improve the detection accuracy and robustness of driving distraction by exploiting the strengths and avoiding the weaknesses. (4. Discussions, paragraph 5).
- Comment: The decision to use simulation, while understandably convenient, raises the question, for EEG in particular, of is the actual driver to wear an eeg headset, or what kind of wearable or remote sensing is needed? Minor corrections are needed to address the practicality issue, in my opinion, which should contain some additional references, noting the use of simulation with EEG to practical purpose, and how that extends to driving in implementation. If this is explainable by the authors, in the article discussion and perhaps other portions of the paper, then the paper would offer a very significant contribution, in laying out a practical method, using new analytical methods, of utilizing EEG for driving monitoring. In doing so, citations of how such an application has worked previously in research or practice is suggested, indicating in detail how such prior research either directly demonstrates that practical value with real-life driving comparison, or alternatively, at least a theoretical analysis of how simulation research with EEG, and EEG in particular, could be altered in practicality or engineering methodology, to allow for practical driver use during real-life driving.
Response: We are very grateful to the reviewer for the suggestions. Based on the suggestion, we added a discussion section, consisting of 5 paragraphs, to the revised manuscript. Paragraph 1 discusses why and how we manually select EEG features. Paragraph 2 discusses why we compare commonly used traditional machine learning classification methods. Paragraph 3 discusses the existing research results, and the problems. Paragraph 4 discusses the advantages of our research method. Paragraph 5 discusses the limitations of this research and future research directions.
Feature extraction and classification are two key techniques for EEG decoding. Feature extraction of EEG signals is a prerequisite for their classification. The main traditional machine learning feature extraction methods for EEG signals are time-domain, frequency-domain, time-frequency-domain, and statistical-feature-based methods. One or more combinations of features of the signal can be selected, e.g., using singular value decomposition (SVD) [28], and wavelet analysis[35] to extract driver EEG features for driving state detection. In our study, the theta, alpha, and beta activities of driver EEG signals during distracted versus focused driving were significantly different based on the fact that the activities of these three frequency bands differed from each other in different brain regions for different types of distracted driving. Therefore, extracting the power spectrum features of the theta, α and β frequency bands of the EEG signals enables the classification of different driving attention states. We hypothesise that this method of extracting power spectral features of only three frequency bands, θ, α and β, will be better compared to the method of extracting power spectral features of the whole frequency band of the EEG signal. Because this method removes the useless features and strengthens the feature differences to make them more separable, the experimental results prove our hypothesis. (4. Discussions, paragraph 1).
Traditional machine learning classification methods for driving distracted EEG signals are Support Vector Machines (SVM) [36], Bayes Classifier[37], k-Nearest Neighbor (kNN) [38], Decision Tree (DT) [39], and Artificial Neural Network (ANN) [40], among others. To find out the classification methods that can classify the extracted EEG power spectrum features well, we compared six methods, namely DT, Discriminant Analysis, Naive Bayes, SVM, kNN, and ANN, the classification performance of these six methods is found to be the best for SVM. (4. Discussions, paragraph 2).
Aiming at the problem that feature extraction of traditional machine learning EEG decoding methods requires manual experience or a priori knowledge, and the decoding accuracy is limited by the low signal-to-noise ratio and spatial resolution of the EEG signals, some researchers have used a pure data approach to implement an end-to-end deep learning method to decode driving distraction EEG signals. For example, CNN is used to implement driving load detection[41], RNN is used to implement cognitive distraction detection [42, 43], and LSTM[44] and GRU[30] are used to detect distraction. However, these methods require a large amount of data to train the model and suffer from overfitting and poor inter-subject robustness. One solution idea is to fuse EEG with other features to improve the accuracy of driving distraction detection, such as using sliding window multiscale entropy (MSE) and bi-directional long and short-term memory network (BiLSTM), sliding window MSE to extract EEG features to determine the location of the distraction, and then statistical analysis of vehicle behavioural data, and finally using BiLSTM to combine MSE and other conventional features to detect driver distraction, and achieved better detection results[45]. A multi-information fusion of electroencephalogram (EEG), electrocardiogram (ECG), electromyogram (EMG), and electrooculogram (EOG)) and behavioral information (PERCLOS70-80-90%, mouth aspect ratio (MAR), eye aspect ratio (EAR), blinking frequency (BF), and head-to-head ratio (HT-R)) was used to detect distraction, firstly, by using a recurrent neural network and a long- and short-term memory ( RNN-LSTM) models to extract physiological features. After fusing these features, classification was performed using a deep residual neural network (DRNN), achieving a detection accuracy of 96.5% [46]. However, the multi-sensor fusion method also suffers from the problem of data synchronisation and is not easy for practical engineering applications. Another solution idea is to combine two or more network models and give full play to the advantages of each network to improve the detection accuracy. For example, combining CNN with GRU to extract the spatio-temporal features of EEG signals, taking advantage of GRU's expertise in solving sequential problems, the accuracy of two-classification (distracted and non-distracted) ranges from 82% to 100%, and the accuracy of three-classification (mobile phone operation distracted, 2-back distracted and non-distracted) ranges from 76% to 98% [30]. Combining LSTM with EEGNet for triple classification of visual, auditory and focused driving achieved an average accuracy of 71.1% [29]. (4. Discussions, paragraph 3).
Our study innovatively combines the provision of EEG features based on manual experience with an improved decoding algorithm. On the one hand, based on the results of existing studies, the initial selection of EEG features is made, which is different from the conventional practice of extracting the full-band features of the EEG, and is replaced by the extraction of the power spectrum features of the θ, α, and β bands, which are the most distinctive and separable features. On the other hand, the combination of GRU and EEGNet is used to form three GRU-EEGNet networks to further feature extraction and classification of the extracted power spectrum features in the θ, α and β bands, respectively. The experimental results prove that the method achieves better results compared with existing similar methods. (4. Discussions, paragraph 4).
Although the method proposed in this paper achieves better results, there are some limitations in this study.1) The experiments in this study are based on a driving simulator, which is completed in an ideal driving environment with no traffic flow. Although the online effect is verified, the validity of the experimental results has not been tested by real vehicle experiments in an actual open-road environment. 2) For example, to be more in line with real driving road conditions, we used a road with many curves and slopes, which would cause the driver to frequently adjust the steering wheel and step on the accelerator and brake pedal, and the frequent body movements would interfere with the EEG signals, and we were not able to filter out these artifacts completely, even though we used a variety of methods. In the next step of the study, 1) use the EEG acquisition equipment in this study and drive a real car to conduct experiments in open road scenarios to test the effectiveness of the method; 2) continue to improve the EEG decoding algorithm to improve detection accuracy and robustness; 3) downsize the EEG electrode channels and use dry electrodes to make the EEG acquisition wireless, portable, and wearable, as wearing a hat is convenient, to facilitate engineering applications and promotion; 4) Combine with other features, such as fusion of multi-sensor information such as camera, EMG, and in-vehicle driving performance sensors, to improve the detection accuracy and robustness of driving distraction by exploiting the strengths and avoiding the weaknesses. (4. Discussions, paragraph 5).

Reviewer 2 Report
Comments and Suggestions for Authors
This study was conducted to assess the efficacy of EEG-based methodologies in detecting driving distraction, contrasting distracted versus focused driving through analysis of theta, alpha, and beta band power spectra. Various models including SVM, EEGNet, and GRU-EEGNet were employed to discern attentional states, with findings indicating superior accuracy of EEGNet over SVM, and GRU-EEGNet surpassing EEGNet in performance. Although the study has its merits, the study rationale appears not well justified, the methods are not well communicated, and the reporting of the results should be significantly improved. Significant revision should be made to ensure the quality and rigor of the study. I hope my comments below will be helpful to the current manuscript further.
Overall
1. The writing should be revised to improve its organization.
2. The purpose in the abstract and the introduction is not clearly stated.
3. What research questions this study is going to address? What’s the theoretical and practical contribution of this study?
Specific comments:
4. In the introduction section, it is important to provide corresponding references for any data and evidence mentioned in the manuscript. Please supplement relevant references to support your argument.
5. The introduction section does not make the research gap evident. A rationale for the necessity of this job must be given.
6. There are some problems with the writing format of the figures and tables. Ensure figures are appropriately titled and numbered, with clear labels and captions that effectively convey the data and its relevance to the study
7. There is no clear comparison between the acquired and previous outcomes. These kinds of comparisons are necessary to verify the legitimacy of the work.
8. The absence of a dedicated discussion section is notable. Such a section is crucial for exploring how the findings contribute to existing literature on driving attention state detection.
9. It should not only discuss the implications of the results but also highlight the innovation and practical implications of the research, thereby situating it within the broader context of human-computer interaction and automotive safety.
10. The references require careful attention to formatting and accuracy. Verify all references for completeness and correctness, ensuring they align with the specified guidelines.
Comments on the Quality of English LanguageExtensive editing of English language required
Author Response
- Comment: The writing should be revised to improve its organization.
Response: Special thanks to the reviewer for the suggestions. As suggested, I have freshened up the introductory section, rewritten the introduction, and added a discussion section.
The act of driving is a complex task that requires the coordinated use of a range of cognitive skills, including planning, memory, motor control, and visual abilities. These abilities vary considerably from one individual to another and depend on cognitive skills and attention levels [1]. Distracted attention reduces the driver's level of perception and decision-making ability, negatively affects vehicle control, and is a significant factor in a large number of accidents[2]. The National Highway Traffic Safety Administration (NHTSA) reports that a total of 39,221 fatal crashes occurred in the United States in FY2022, involving 60,048 drivers and resulting in 42,514 deaths. Of these, 3,047 crashes were related to driver distraction, accounting for 8% of all fatalities[3]. Several studies have demonstrated that distraction while driving impairs driving performance[4-13]. If driving distraction can be detected in a timely and effective manner and drivers can be intervened, the degree of injury or even the accident can be reduced or even avoided. Therefore, the research on driving distraction detection methods is of great significance to improve driving safety. (1. Introduction, paragraph 1).
There are five common methods for detecting driving distractions: performance-based, driver body posture, driver physiological characteristics, subjective report, and hybrid detection[14]. However, all of these methods have certain limitations; it is difficult to distinguish between types of distraction in driving performance-based methods, the use of a camera to analyze the driver's body posture to detect distraction is susceptible to light (e.g., daytime and nighttime), the subjective report is subject to subjectivity, and the mixed measurements suffer from the problem of difficulty in synchronizing the data from multiple data sources. EEG, on the other hand, is the result of a series of drivers' mental activities and is characterized by objective accuracy, high temporal resolution, and the ability to detect different types of distraction. In particular, compared with other methods that can only detect distraction when it has already occurred or after it has occurred, EEG technology can be detected when the distraction has just occurred. This is because physiological changes begin when the driver is in the initial stages of distraction. Therefore, driving distraction detection using EEG technology is an effective means to improve driving safety. (1. Introduction, paragraph 2).
Differences in the characteristics of EEG signals in drivers' different driving attention states were found and used to detect distraction. EEG signals theta (4-8 Hz), alpha (8-13 Hz) and beta (13-35 Hz) activities during driving distraction are significantly different from those during focused driving[15]. And this difference varies for different types of distraction. The EEG signal alpha spindle rate was significantly higher during auditory distraction than during focused driving without a subtask[16]. Right parietal alpha and beta activity were negatively correlated with the degree of visual distraction[17]. Alpha and theta band power were positively correlated with cognitive load[18]. This difference has also been demonstrated using neuroimaging to study driving attention states and brain region activation, which found significant activation in the right frontal cortical regions of drivers during distraction[19, 20]. (1. Introduction, paragraph 3).
EEG technology has been widely used for the detection of distracted driving, such as visual distraction[17, 21], auditory distraction[22, 23], and cognitive distraction[24-27], due to its ease of portability, insensitivity to head motion, and high temporal resolution. However, EEG also has the characteristic of a low signal-to-noise ratio due to the signal being susceptible to environmental noise, as well as physiological activity and motion artifacts, which are difficult to eliminate, resulting in poor accuracy of distraction detection. Therefore, current research focuses on EEG decoding algorithms to improve the accuracy of distraction detection. Some studies used traditional machine learning algorithms for EEG decoding, such as using singular value decomposition (SVD) to extract the features of each frequency band of the EEG signal, and then using SVM to classify them to achieve the detection of the driver's cognitive state, which achieved a detection accuracy of 96.78%[28]. Other studies used deep learning methods for decoding, such as Yan et al [29] used EEGNet combined with LSTM to detect visual and auditory distraction, and obtained an average accuracy of 71.1%. Li et al[30]used CNN combined with GRU (Gate Recurrent Unit) for dichotomous classification (distracted vs. non-distracted), and the accuracy between 82%-100% and triple classification (mobile phone operation distraction, 2-back distraction and non-distraction) between 76%-98%.Wang et al [31]performed driver cognitive load detection by extracting the EEG signal θ, α, β, and γ band power spectra by using neural network and SVM, respectively, and the best model achieved a classification accuracy of 90.37%. These methods are still some way from practical engineering applications due to the limitation of low detection accuracy, after all, frequent false warnings in the driving process will cause disturbances to the driver and are not conducive to driving safety. (1. Introduction, paragraph 4).
Our study adopts a combination of manual selection of EEG features and deep learning to give full play to each other's advantages, aiming to effectively improve the detection accuracy. On the one hand, the performance of the EEG decoding algorithm is improved, i.e., GRU is combined with EEGNet to form a GRU-EEGNet network; on the other hand, based on the characteristics of different power of the driver's EEG signals in the θ, α, and β frequency bands when driving with distraction and driving with concentration, we firstly extracted the power features of the θ, α, and β frequency bands of the EEG signals, and then used three GRU-EEGNet models to decode the θ, α and β band power features are then decoded using three GRU-EEGNet models. We have three hypotheses: one using the method of extracting only the power spectral features of the EEG signal θ, α and β bands is better than the method of extracting the whole features of the five bands of the EEG signal, two using the EEGNet model method is better than the SVM for classification, and three the improved GRU-EEGNet has a better classification effect than the EEGNet model. We use a Logitech G29 driving simulator to carry out distracted driving experiments, the experimental scene and distraction sub-tasks are constructed by Unity3D. The driver's EEG signals were synchronously acquired during the experiment, and the acquired EEG signals were pre-processed and processed separately. By extracting the theta, α and β band power spectral features of the driver's EEG signals, the SVM and EEGNet models and the GRU-EEGNet method were then used to detect the driving attention state, respectively. (1. Introduction, paragraph 5).
Feature extraction and classification are two key techniques for EEG decoding. Feature extraction of EEG signals is a prerequisite for their classification. The main traditional machine learning feature extraction methods for EEG signals are time-domain, frequency-domain, time-frequency-domain, and statistical-feature-based methods. One or more combinations of features of the signal can be selected, e.g., using singular value decomposition (SVD) [28], and wavelet analysis[35] to extract driver EEG features for driving state detection. In our study, the theta, alpha, and beta activities of driver EEG signals during distracted versus focused driving were significantly different based on the fact that the activities of these three frequency bands differed from each other in different brain regions for different types of distracted driving. Therefore, extracting the power spectrum features of the theta, α and β frequency bands of the EEG signals enables the classification of different driving attention states. We hypothesise that this method of extracting power spectral features of only three frequency bands, θ, α and β, will be better compared to the method of extracting power spectral features of the whole frequency band of the EEG signal. Because this method removes the useless features and strengthens the feature differences to make them more separable, the experimental results prove our hypothesis. (4. Discussions, paragraph 1).
Traditional machine learning classification methods for driving distracted EEG signals are Support Vector Machines (SVM) [36], Bayes Classifier[37], k-Nearest Neighbor (kNN) [38], Decision Tree (DT) [39], and Artificial Neural Network (ANN) [40], among others. To find out the classification methods that can classify the extracted EEG power spectrum features well, we compared six methods, namely DT, Discriminant Analysis, Naive Bayes, SVM, kNN, and ANN, the classification performance of these six methods is found to be the best for SVM. (4. Discussions, paragraph 2).
Aiming at the problem that feature extraction of traditional machine learning EEG decoding methods requires manual experience or a priori knowledge, and the decoding accuracy is limited by the low signal-to-noise ratio and spatial resolution of the EEG signals, some researchers have used a pure data approach to implement an end-to-end deep learning method to decode driving distraction EEG signals. For example, CNN is used to implement driving load detection[41], RNN is used to implement cognitive distraction detection [42, 43], and LSTM[44] and GRU[30] are used to detect distraction. However, these methods require a large amount of data to train the model and suffer from overfitting and poor inter-subject robustness. One solution idea is to fuse EEG with other features to improve the accuracy of driving distraction detection, such as using sliding window multiscale entropy (MSE) and bi-directional long and short-term memory network (BiLSTM), sliding window MSE to extract EEG features to determine the location of the distraction, and then statistical analysis of vehicle behavioural data, and finally using BiLSTM to combine MSE and other conventional features to detect driver distraction, and achieved better detection results[45]. A multi-information fusion of electroencephalogram (EEG), electrocardiogram (ECG), electromyogram (EMG), and electrooculogram (EOG)) and behavioral information (PERCLOS70-80-90%, mouth aspect ratio (MAR), eye aspect ratio (EAR), blinking frequency (BF), and head-to-head ratio (HT-R)) was used to detect distraction, firstly, by using a recurrent neural network and a long- and short-term memory ( RNN-LSTM) models to extract physiological features. After fusing these features, classification was performed using a deep residual neural network (DRNN), achieving a detection accuracy of 96.5% [46]. However, the multi-sensor fusion method also suffers from the problem of data synchronisation and is not easy for practical engineering applications. Another solution idea is to combine two or more network models and give full play to the advantages of each network to improve the detection accuracy. For example, combining CNN with GRU to extract the spatio-temporal features of EEG signals, taking advantage of GRU's expertise in solving sequential problems, the accuracy of two-classification (distracted and non-distracted) ranges from 82% to 100%, and the accuracy of three-classification (mobile phone operation distracted, 2-back distracted and non-distracted) ranges from 76% to 98% [30]. Combining LSTM with EEGNet for triple classification of visual, auditory and focused driving achieved an average accuracy of 71.1% [29]. (4. Discussions, paragraph 3).
Our study innovatively combines the provision of EEG features based on manual experience with an improved decoding algorithm. On the one hand, based on the results of existing studies, the initial selection of EEG features is made, which is different from the conventional practice of extracting the full-band features of the EEG, and is replaced by the extraction of the power spectrum features of the θ, α, and β bands, which are the most distinctive and separable features. On the other hand, the combination of GRU and EEGNet is used to form three GRU-EEGNet networks to further feature extraction and classification of the extracted power spectrum features in the θ, α and β bands, respectively. The experimental results prove that the method achieves better results compared with existing similar methods. (4. Discussions, paragraph 4).
Although the method proposed in this paper achieves better results, there are some limitations in this study.1) The experiments in this study are based on a driving simulator, which is completed in an ideal driving environment with no traffic flow. Although the online effect is verified, the validity of the experimental results has not been tested by real vehicle experiments in an actual open-road environment. 2) For example, to be more in line with real driving road conditions, we used a road with many curves and slopes, which would cause the driver to frequently adjust the steering wheel and step on the accelerator and brake pedal, and the frequent body movements would interfere with the EEG signals, and we were not able to filter out these artifacts completely, even though we used a variety of methods. In the next step of the study, 1) use the EEG acquisition equipment in this study and drive a real car to conduct experiments in open road scenarios to test the effectiveness of the method; 2) continue to improve the EEG decoding algorithm to improve detection accuracy and robustness; 3) downsize the EEG electrode channels and use dry electrodes to make the EEG acquisition wireless, portable, and wearable, as wearing a hat is convenient, to facilitate engineering applications and promotion; 4) Combine with other features, such as fusion of multi-sensor information such as camera, EMG, and in-vehicle driving performance sensors, to improve the detection accuracy and robustness of driving distraction by exploiting the strengths and avoiding the weaknesses. (4. Discussions, paragraph 5).
- Comment: The purpose in the abstract and the introduction is not clearly stated.
Response: We are very grateful to the reviewer for the suggestions. As suggested, the introductory section was rewritten, in which the purpose of the study was elaborated, and the abstract was modified accordingly. Please see the response to Comment 1.
The extraction of the θ, α and β band power spectrum features of the EEG signals was found to be a more effective method than the extraction of the power spectrum features of the whole EEG signals for the detection of driving attention states. The driving attention state detection accuracy of the proposed GRU-EEGNet model is improved by 6.3% and 12.8% over the EEGNet model and PSD_SVM method, respectively. It is shown that the EEG decoding method combining EEG features and an improved deep learning algorithm is manually and preliminarily selected based on the results of existing studies, which effectively improves the driving attention state detection accuracy. (Abstract, line 6-12).
- Comment: What research questions this study is going to address? What’s the theoretical and practical contribution of this study?
Response: Special thanks to the reviewer for the suggestion. In the rewritten introductory section and the added discussion section, the problems, theoretical and practical contributions to be addressed by this study are systematically described. Please see the response to Comment 1.
- Comment: In the introduction section, it is important to provide corresponding references for any data and evidence mentioned in the manuscript. Please supplement relevant references to support your argument.
Response: Thanks to the reviewer for the proposed suggestion. In the introduction section of the original manuscript, the data related to traffic accidents in 2021 published on the official website of NHTSA was quoted, but no reference was provided, and in the revised manuscript, we quoted the latest data related to traffic accidents in 2022 published on the official website of NHTSA, and a reference was provided.
The National Highway Traffic Safety Administration (NHTSA) reports that a total of 39,221 fatal crashes occurred in the United States in FY2022, involving 60,048 drivers and resulting in 42,514 deaths. Of these, 3,047 crashes were related to driver distraction, accounting for 8% of all fatalities[3].(1. Introduction, paragraph 1, line 6-9).
- Comment: The introduction section does not make the research gap evident. A rationale for the necessity of this job must be given.
Response: Special thanks to the reviewer for the suggestion. This is systematically presented and discussed in a rewritten introduction and a new discussion section. Please see the response to Comment 1.
Our study adopts a combination of manual selection of EEG features and deep learning to give full play to each other's advantages, aiming to effectively improve the detection accuracy. On the one hand, the performance of the EEG decoding algorithm is improved, i.e., GRU is combined with EEGNet to form a GRU-EEGNet network; on the other hand, based on the characteristics of different power of the driver's EEG signals in the θ, α, and β frequency bands when driving with distraction and driving with concentration, we firstly extracted the power features of the θ, α, and β frequency bands of the EEG signals, and then used three GRU-EEGNet models to decode the θ, α and β band power features are then decoded using three GRU-EEGNet models. (1. Introduction, paragraph 5, line 1-8).
- Comment: There are some problems with the writing format of the figures and tables. Ensure figures are appropriately titled and numbered, with clear labels and captions that effectively convey the data and its relevance to the study.
Response: We are very grateful to the reviewer’s suggestions. We have amended the figure names for Figures 1, 5, 7, 9, 10, 11 and 12.
- Comment: There is no clear comparison between the acquired and previous outcomes. These kinds of comparisons are necessary to verify the legitimacy of the work.
Response: The reviewer’s comments were very insightful and meaningful. In the added discussion section, the results of currently available research are presented and compared with the methodology of this study.
Aiming at the problem that feature extraction of traditional machine learning EEG decoding methods requires manual experience or a priori knowledge, and the decoding accuracy is limited by the low signal-to-noise ratio and spatial resolution of the EEG signals, some researchers have used a pure data approach to implement an end-to-end deep learning method to decode driving distraction EEG signals. For example, CNN is used to implement driving load detection[41], RNN is used to implement cognitive distraction detection [42, 43], and LSTM[44] and GRU[30] are used to detect distraction. However, these methods require a large amount of data to train the model and suffer from overfitting and poor inter-subject robustness. One solution idea is to fuse EEG with other features to improve the accuracy of driving distraction detection, such as using sliding window multiscale entropy (MSE) and bi-directional long and short-term memory network (BiLSTM), sliding window MSE to extract EEG features to determine the location of the distraction, and then statistical analysis of vehicle behavioural data, and finally using BiLSTM to combine MSE and other conventional features to detect driver distraction, and achieved better detection results[45]. A multi-information fusion of electroencephalogram (EEG), electrocardiogram (ECG), electromyogram (EMG), and electrooculogram (EOG)) and behavioral information (PERCLOS70-80-90%, mouth aspect ratio (MAR), eye aspect ratio (EAR), blinking frequency (BF), and head-to-head ratio (HT-R)) was used to detect distraction, firstly, by using a recurrent neural network and a long- and short-term memory ( RNN-LSTM) models to extract physiological features. After fusing these features, classification was performed using a deep residual neural network (DRNN), achieving a detection accuracy of 96.5% [46]. However, the multi-sensor fusion method also suffers from the problem of data synchronisation and is not easy for practical engineering applications. Another solution idea is to combine two or more network models and give full play to the advantages of each network to improve the detection accuracy. For example, combining CNN with GRU to extract the spatio-temporal features of EEG signals, taking advantage of GRU's expertise in solving sequential problems, the accuracy of two-classification (distracted and non-distracted) ranges from 82% to 100%, and the accuracy of three-classification (mobile phone operation distracted, 2-back distracted and non-distracted) ranges from 76% to 98% [30]. Combining LSTM with EEGNet for triple classification of visual, auditory and focused driving achieved an average accuracy of 71.1% [29]. (4. Discussions, paragraph 3).
Our study innovatively combines the provision of EEG features based on manual experience with an improved decoding algorithm. On the one hand, based on the results of existing studies, the initial selection of EEG features is made, which is different from the conventional practice of extracting the full-band features of the EEG, and is replaced by the extraction of the power spectrum features of the θ, α, and β bands, which are the most distinctive and separable features. On the other hand, the combination of GRU and EEGNet is used to form three GRU-EEGNet networks to further feature extraction and classification of the extracted power spectrum features in the θ, α and β bands, respectively. The experimental results prove that the method achieves better results compared with existing similar methods. (4. Discussions, paragraph 4).
- Comment: The absence of a dedicated discussion section is notable. Such a section is crucial for exploring how the findings contribute to existing literature on driving attention state detection.
Response: Thanks to the reviewer for the very constructive suggestion. As suggested, we have added a new discussion section to the revised manuscript. Please see the response to Comment 1.
- Comment: It should not only discuss the implications of the results but also highlight the innovation and practical implications of the research, thereby situating it within the broader context of human-computer interaction and automotive safety.
Response: Thank you for bringing this matter into our attention. The innovativeness of this paper is discussed in paragraph 4 of the newly added discussion section, and the practicality of the methodology of this research is prospected in paragraph 5.
Our study innovatively combines the provision of EEG features based on manual experience with an improved decoding algorithm. On the one hand, based on the results of existing studies, the initial selection of EEG features is made, which is different from the conventional practice of extracting the full-band features of the EEG, and is replaced by the extraction of the power spectrum features of the θ, α, and β bands, which are the most distinctive and separable features. On the other hand, the combination of GRU and EEGNet is used to form three GRU-EEGNet networks to further feature extraction and classification of the extracted power spectrum features in the θ, α and β bands, respectively. The experimental results prove that the method achieves better results compared with existing similar methods. (4. Discussions, paragraph 4).
Although the method proposed in this paper achieves better results, there are some limitations in this study.1) The experiments in this study are based on a driving simulator, which is completed in an ideal driving environment with no traffic flow. Although the online effect is verified, the validity of the experimental results has not been tested by real vehicle experiments in an actual open-road environment. 2) For example, to be more in line with real driving road conditions, we used a road with many curves and slopes, which would cause the driver to frequently adjust the steering wheel and step on the accelerator and brake pedal, and the frequent body movements would interfere with the EEG signals, and we were not able to filter out these artifacts completely, even though we used a variety of methods. In the next step of the study, 1) use the EEG acquisition equipment in this study and drive a real car to conduct experiments in open road scenarios to test the effectiveness of the method; 2) continue to improve the EEG decoding algorithm to improve detection accuracy and robustness; 3) downsize the EEG electrode channels and use dry electrodes to make the EEG acquisition wireless, portable, and wearable, as wearing a hat is convenient, to facilitate engineering applications and promotion; 4) Combine with other features, such as fusion of multi-sensor information such as camera, EMG, and in-vehicle driving performance sensors, to improve the detection accuracy and robustness of driving distraction by exploiting the strengths and avoiding the weaknesses. (4. Discussions, paragraph 5).
- Comment: The references require careful attention to formatting and accuracy. Verify all references for completeness and correctness, ensuring they align with the specified guidelines.
Response: Thank you for your question. We have verified and corrected all references.
